# Application of deep learning algorithm to detect and visualize vertebral fractures on plain frontal radiographs

**Hsuan-Yu Chen**[1,2,3], **Benny Wei-Yun Hsu**[4], **Yu-Kai Yin**[4], **Feng-Huei Lin**[1], **Tsung-Han Yang**[2,3], **Rong-Sen Yang**[2], **Chih-Kuo Lee**[5], **Vincent S. Tseng**[4]*

1 Institute of Biomedical Engineering, National Taiwan University, Taipei City, Taiwan, **2** Department of Orthopedics, National Taiwan University College of Medicine and National Taiwan University Hospital, Taipei, Taiwan, **3** Department of Orthopedics, National Taiwan University HsinChu Hospital, Hsin-Chu, Taiwan, **4** Institute of Computer Science and Engineering, National Chiao Tung University, Hsinchu, Taiwan, **5** Department of Internal Medicine, National Taiwan University HsinChu Hospital, HsinChu, Taiwan

* vtseng@cs.nctu.edu.tw

**Data Availability Statement:** There are ethical restrictions being placed upon the image data, for image data contain potentially identifying or sensitive patient information and Institutional

## Abstract

### Background

Identification of vertebral fractures (VFs) is critical for effective secondary fracture prevention owing to their association with the increasing risks of future fractures. Plain abdominal frontal radiographs (PARs) are a common investigation method performed for a variety of clinical indications and provide an ideal platform for the opportunistic identification of VF. This study uses a deep convolutional neural network (DCNN) to identify the feasibility for the screening, detection, and localization of VFs using PARs.

### Methods

A DCNN was pretrained using ImageNet and retrained with 1306 images from the PARs database obtained between August 2015 and December 2018. The accuracy, sensitivity, specificity, and area under the receiver operating characteristic curve (AUC) were evaluated. The visualization algorithm gradient-weighted class activation mapping (Grad-CAM) was used for model interpretation.

### Results

Only 46.6% (204/438) of the VFs were diagnosed in the original PARs reports. The algorithm achieved 73.59% accuracy, 73.81% sensitivity, 73.02% specificity, and an AUC of 0.72 in the VF identification.

### Conclusion

Computer driven solutions integrated with the DCNN have the potential to identify VFs with good accuracy when used opportunistically on PARs taken for a variety of clinical purposes. The proposed model can help clinicians become more efficient and economical in the current clinical pathway of fragile fracture treatment.

Review Board has imposed them. Data inquiries can be made to the following: Institution Review Board of National Taiwan University Hospital Hsin-Chu Branch Contact information: Tel:886-3-5326151#8665 Fax:886-3-5333568 Email: cychen1@hch.gov.tw.

**Funding:** The author(s) received no specific funding for this work.

**Competing interests:** NO authors have competing interests.

# Introduction

Vertebral fractures (VFs), which are identified as deformities of vertebral bodies based on the imaging of the lateral spine, are a hallmark of osteoporosis. VFs are the most prevalent fractures of all, although they remain largely undiagnosed; this is because VFs are asymptomatic or induce only a mild pain and there is a lack of routine radiographic detection in the clinical pathway [1]. In Europe, the economic burden, which is estimated at € 37 billion, is expected to increase by 25% by 2025 [2]. Plain abdominal frontal radiographs (PARs) are common investigation methods performed for a variety of clinical conditions, such as urinary or gastrointestinal disorders, and it is challenging for physicians to screen VFs based on PARs. Convolutional neural networks (CNN) [3] are capable of processing data in the form of images, videos, signals, sequences, and so on. In the architecture of a CNN, each convolution process is executed by a filter, which extracts the local information from the different parts of an image. The CNN can obtain local information from many different levels and combine these extracted features to build the global information. Numerous researchers in the field of medical imaging have been using CNN-based models to accomplish various tasks, and their successes in medicine provide a myriad of potential prospects to the deep learning research [4–7]. Many methods such as gradient-weighted class activation mapping (Grad-CAM) [8] have been developed for the visual depiction of a deep convolutional neural network (DCNN) to assist clinicians in identifying the pathologic regions and validate the performance of DCNNs. Furthermore, PARs provide an ideal platform for the opportunistic identification of VFs since general physicians typically focus their practice on certain disease categories. The automated VF diagnosis algorithm, which is trained based on the DCNN, has the potential for increasing efficiency, reducing delayed management, and improving patient outcomes, especially those in need of osteoporotic treatment but have not been diagnosed.

# Materials and methods

## Study population

A total of 1456 PARs, obtained between 2015 and 2018 from a database developed during the Fracture Liaison Services (FLS) program of the National Taiwan University Hospital (NTUH), Hsin-Chu branch, were retrospectively enrolled in the study. All medical history, demographic data, imaging findings, follow-up material, and complications were recorded in this database. The study protocol was approved by the institutional review board (IRB No.: 108-007-E) of the National Taiwan University Hospital, Hsin-Chu Branch on March 01, 2019 and the IRB waived the need for consent.

## Vertebral fracture assessment

VFs are defined as a vertebral body height loss of 20% or more in either the anterior, middle, or posterior vertebral body height by comparing the baseline and the closeout on lateral thoraco-lumbar spinal radiographs. This is accomplished using the semi-quantitative method of Genant, which classified VFs as grade I (mild, height loss between 20%–25%), grade II (moderate, height loss between 25%–40%), and grade III (severe, height loss over 40%) [9].

## Image labeling and database establishment

Images using a picture archiving and communication system (PACS) viewer were stored using a Python script. The color was 8-bit grayscale and the size of the stored images varied from 2128 x 2248 pixels to 2688 x 2688 pixels. All the PARs included in our study contained their original radiology reports and lateral spinal radiographs. The PARs datasets were initially labeled as VF or non-VF according to the diagnosis in the registry, and supportive images

such as lateral thoraco-lumbar plain film, CT, MRI, or other related images were reviewed and reported, respectively, by one radiologist and one spine surgeon. The diagnosis was finalized when the two observers agreed with each other. One hundred and fifty PARs with poor-quality images—such as poor image contrast, positioning errors, or the presence of a foreign body—and those with VFs due to high-energy trauma, spine deformity, metastatic tumor, infection, tuberculosis, Scheurmann's disease, scoliosis, or previous spine surgery, were excluded based on the medical history and/or observations made from the radiographs.

## Development of the algorithm

In recent years, many CNN-based approaches have been developed to solve classification problems. However, no CNN-based method that can be applied to all types of image classification problem has yet been developed. Each CNN-based method has its own advantages according to problem definitions and data characteristics. In the various such methods, ResNet [10] has been proven to be one of the most representative deep learning networks. Furthermore, following the architecture of this network, modifications to the architecture of the existing CNN-based methods have been made by utilizing the idea of residual blocks to achieve deeper networks and more efficient learning [11]. There are several deep learning methods that have been developed based on the ResNet architecture. However, as mentioned before, the type of problem and the data characteristics determine the method to be used. In this study, we consider a typical classification method and verify its efficacy. We use ResNeXt [12] as the backbone model for our study. ResNeXt, which is the enhanced network of the ResNet, has shown outstanding results in image classification and has been applied to the field of medical image analysis [13]. ResNeXt uses the concept of cardinality and introduces the multi-branch structure of inception network into its own structure [14]. ResNeXt divides the residual layer into several connections, each connection being identical to the others. Medical data are difficult to obtain and the amount of data is usually insufficient for a model to learn the optimal parameters. In this study, we use ResNeXt with 50 layers (ResNeXt-50, cardinality = 32, channel = 4). Transfer learning [15] provides a highly effective way for the network to learn the parameters by transferring them from one task to the other, and the datasets for these two tasks need not be the same. In the case of image data, a shallow network learns how to capture the general low-level features of objects, such as the edges; in contrast, a deep network learns how to extract the high-level features such as the texture. However, if the dataset is extremely small, the network struggles to learn these features, irrespective of whether the network is shallow or deep. To solve this problem, we employ a network that is trained using a large dataset and has learned a general method of capturing the features. We then apply this network to the target task. More specifically, we apply a shallow network trained using a large dataset to a medical task consisting of limited but valuable medical data. Then, we connect this shallow network to a deep network trained using the same medical data as mentioned above. The resulting network is expected to have better ability to learn both general low- and high-level features. In our study, for the ResNeXt trained using the ImageNet dataset [16], we use shallow layers and freeze the parameters of these layers. The ImageNet is a large repository of various types of images. Although the ImageNet is not a medical specific dataset, it can help us in obtaining better initial parameters of the network during learning [17]. Notably, the shallow layers of the ResNeXt are pretrained, and its deep layers are trained using our PARs database.

## Evaluating the algorithm

Image preprocessing (Fig 1), including image resizing into 224 x 224 pixels, adjusting color jitter, and image normalization (for the purpose of removing noise and avoiding noisy pixel

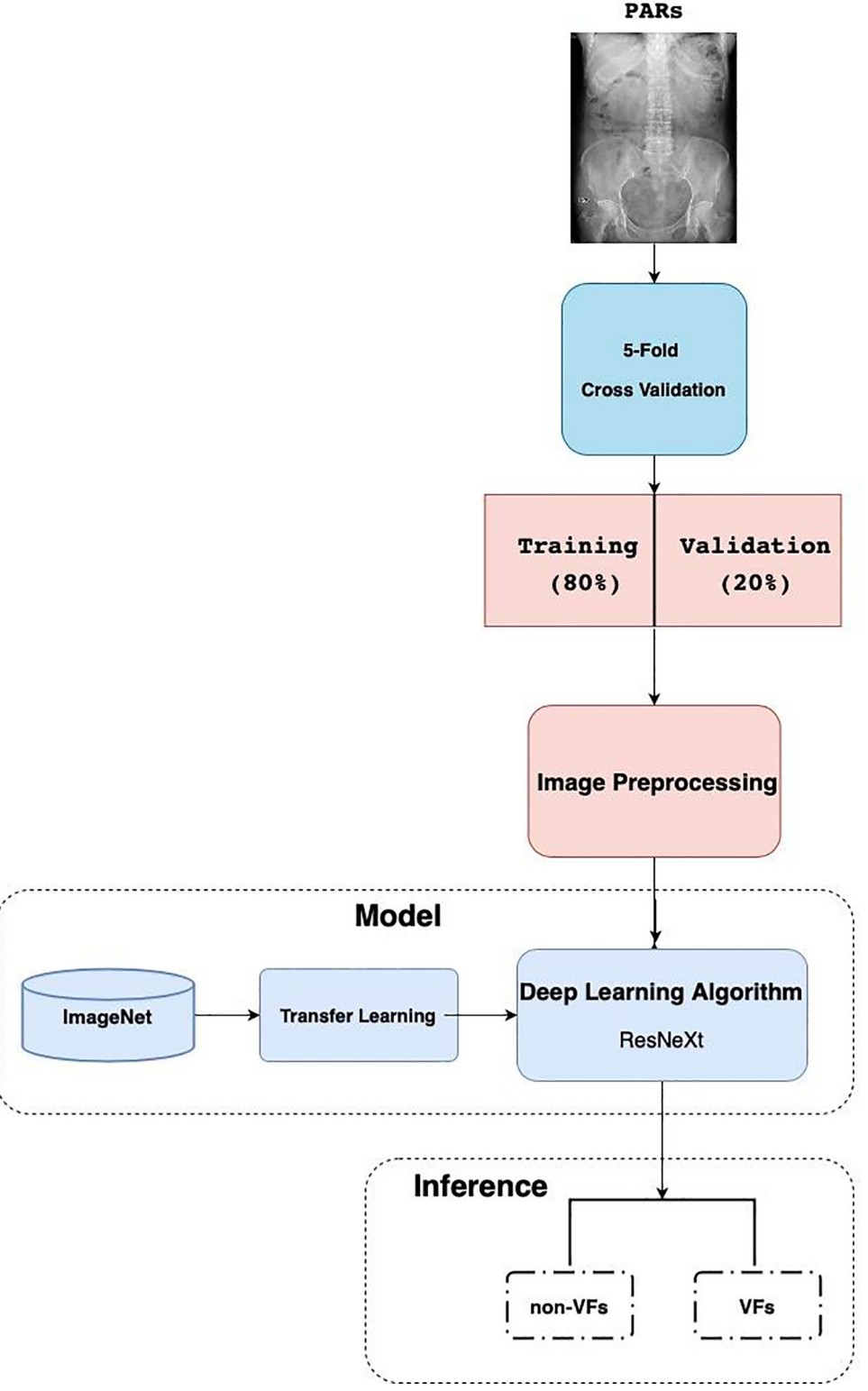

**Fig 1. Data preparation, development, and evaluation of the Deep Convolutional Neural Network (DCNN).**

values from influencing network training) were performed. PARs dataset was split as 80% as training data and 20% as validation data. To ensure the reliability of the experiments, we applied 5-fold cross validation method as our training plan. In each set, the proportion of fracture image and normal image was equal. Every pixel in each image was calculated, and we estimated the mean and variance for the pixel set. ResNeXt was chosen as our basic model architecture in tandem with transfer learning methods. ImageNet—the biggest image dataset in the world containing over 1.2 million images and approximately 1000 classes for training set—was used as a pretraining material for transfer learning to enhance the model performance. The following metrics were used in the training and testing process: 1) accuracy, 2) sensitivity, and 3) specificity.

## Statistical analysis

The software used to build DCNN was based on open-source library with Python 3.7 and PyTorch, and the training process was run on an Intel Core i7-7740X CPU 4.30 GHz with GeForce GTX 1080Ti GPU. DCNN model and clinicians were compared using the sensitivity, specificity, and accuracy metrics. ROC curves and AUCs were used to evaluate the performance of the model.

## Results

We examined 1306 radiologist reports of PARs (VF: non-VF = 438: 868) in our database and only 46.6% (204/438) VFs were diagnosed in the original PARs reports. The original model was trained using only our dataset and showed 61.11% accuracy, 100% sensitivity, and 0% specificity on the validation dataset in the identification of VFs. The model weights were preserved and retrained using the ImageNet. The final results on the validation set were as follows: accuracy of 73.59%, sensitivity of 73.81% sensitivity, and specificity of 73.02%. The change in accuracy and loss during training are shown in Fig 2. The results demonstrate that the pretraining technique helps in achieving a better performance. Ten clinicians, including three orthopedic surgeons, two radiologists, and five physicians, were tested by means of a questionnaire. The mean accuracy was 76.8% (ranging from 71.4% to 87.9%), the mean sensitivity was 76.9% (ranging from 70.5% to 88.6%), and the mean specificity was 76.8% (ranging from 69.2% to 87.4%). The ROC curve of the prediction probability compared with that obtained

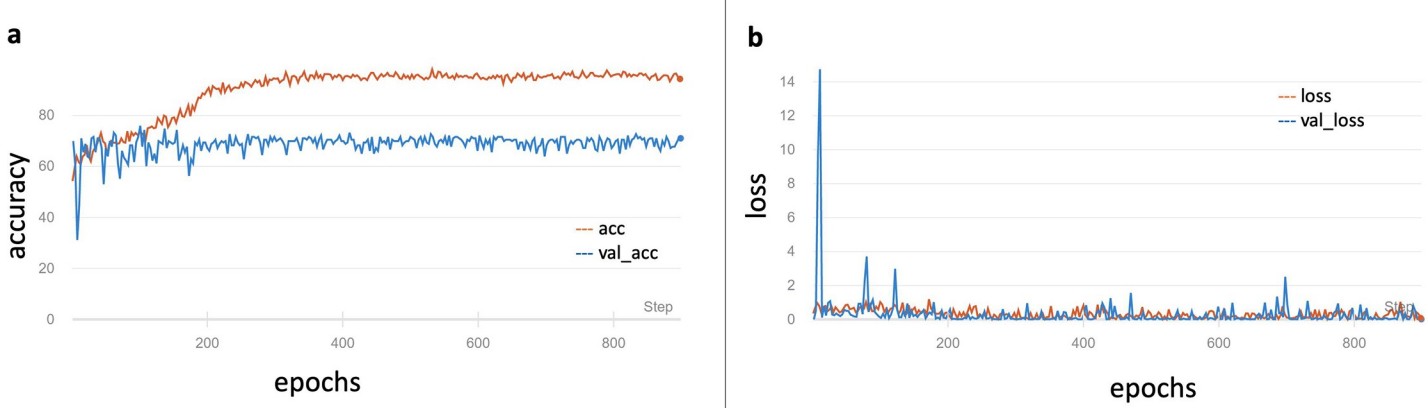

**Fig 2. Performance of DCNN in the training and validation datasets.** a. Accuracy change during training process. b. Change of loss during training process. acc = accuracy of the training set; val_acc = accuracy of the validation set; loss = loss of the training set; val_loss = loss of the validation set.

from experts and the model is shown in Fig 3. The model achieved an AUC of 0.72; thus, the mean performance of clinicians was still better than that of the model.

## Discussion

VFs, asymptomatic or causing mild pain, continue to be a highly under-diagnosed disease, and the increase in its incidence rates is believed to be due to the increase in the elderly population worldwild. Rapid identification of asymptomatic VFs is challenging owing to the limitations of different specialists, who tend to focus on their practice in their own specific fields of expertise.

VFs are a strong predictor of future fractures, which are associated with an increased risk of death, probable chronic back pain, kyphotic deformity, immobility, and a loss of self-esteem. Early detection and adequate treatment are the overriding goals in managing osteoporosis; these goals are critical for patient survival and the preservation of active life, especially in post-menopausal women. The early and accurate diagnosis of vertebral fractures is therefore crucial for the treatment of VFs because it can lead to a 70% reduction in the risk of morphometric VFs, 41% reduction in the risk of hip fractures, and 25% reduction in the risk of other types of fractures [18].

PARs, which are especially insensitive in the diagnosis of VFs, are a convenient platform for the opportunistic identification of VFs but are far from ideal. Endplates must be seen for the diagnosis of VFs, but these endplates, which are not parallel to the X-ray beam on PARs, are often invisible or poorly seen. Doctors cannot diagnose the presence or absence of fractures with confidence if the endplates are not seen. Furthermore, if there is an endplate tilting, the posterior and anterior margins of a particular body may line up in such a way as to falsely suggest VFs on PARs. Rhee et al estimate that 15% of the VFs are not visible on PARs, even when the lateral view is included [19].

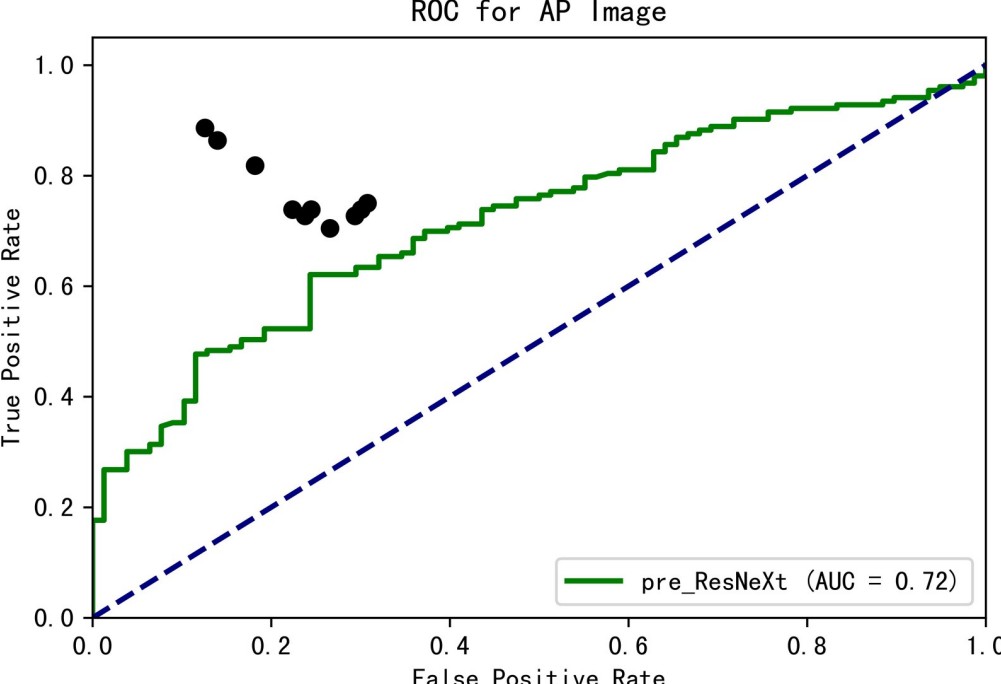

**Fig 3. ROC curve of prediction probability compared with experts and the DCNN.** The black spots indicate the performance of clinicians.

The automated VF diagnosis algorithm is expected to play a greater role in promoting preventive measures against osteoporosis and increasing clinician's awareness. The computer-assisted diagnosis (CAD) system, which has been integrated as part of PACS, combines elements of AI and computer vision with radiological image processing to improve efficiency in daily clinical practice [20]. Automation of bony fracture detection has been discussed in several studies; however, VFs diagnosis remains a challenge not only when humans have to perform the diagnosis but also when it needs to be integrated with artificial intelligence [21, 22]. Detection of subclinical or undiagnosed VFs on routine PARs has been shown to have the capability to identify patients who are at triple risks of developing a future hip fracture. The FLS program performed poorly in identifying VFs, and the national and international guidelines encouraged opportunistic detection. Furthermore, owing to the availability of near-perfect ground truth labels, VFs are now a promising target for deep learning approaches. Most models developed previously for VFs were region-based and required a local segmentation network to first identify the vertebrae. Our study shows that the detection and diagnosis of VFs on PARs could be performed without the segmentation of each vertebrae by extracting domain-specific visual features from the input of a whole-scale radiograph. A similar accuracy level was achieved between the deep learning algorithm and the clinicians. Our DCNN takes PARs and detects the presence of VFs automatically. The results of this study demonstrate that DCNNs can be trained without specifying the lesion-based features; this can be done by using sizable non-pixel labeled datasets. The time required for manual segmentation and labeling can thus be saved. However, ensuring superior performance using deep learning systems requires the availability of specific radiographic data and the development of large, clean datasets. We used 5-fold cross validation and transfer learning from pretrained datasets to meet these requirements. For transfer learning, we performed two types of methods: 1) Freezing of the parameters for the first module in the basic model; 2) Freezing of the parameters for the last module in the basic model. Notably, high-level semantic features should be learned from the PARs database, and if we replace the high-level semantic features with the parameters learned from ImageNet data, irrelevant results are obtained. We applied the transfer learning method and pretrained model using ImageNet images. The resulting accuracy, consequently, increased from 61.11% (scratch pretrained) to 73.59%, and the pretraining material impacted the final accuracy.

However, there are some limitations of our study. One fundamental limitation is that the nature of DCNNs for the system was afforded only images and the associated diagnosis without explicit definitions of the features. The "black box" mechanism is one paradox in DCNNs concerned with the analysis of medical images; this is because the logic process followed by these networks is different between those of humans and computers. Deep learning may use data characteristics previously unknown to or ignored by humans owing to the different logic processes used by them, (unless, it is just a crude guess). Although promising results were obtained in our study, the exact features that were used are unknown. Next, our experiments, an overfitting problem was encountered during modeling. There are two ways to address this problem in the future. One involves the collection of additional data to make the dataset more diverse. The other involve the reduction in the number of parameters of the deep learning model. A deep learning model usually consists of millions of parameters that are used for data fitting. However, such high numbers of parameters may cause the model to only focus on the training data. The model loses the capability of classifying the unknown data.

Furthermore, giving the entire image as an input to the model is also challenging. To this end, we design a software to eliminate the unnecessary information, such as patient background data and the non-uniform range of the plain films. We only intend to include those

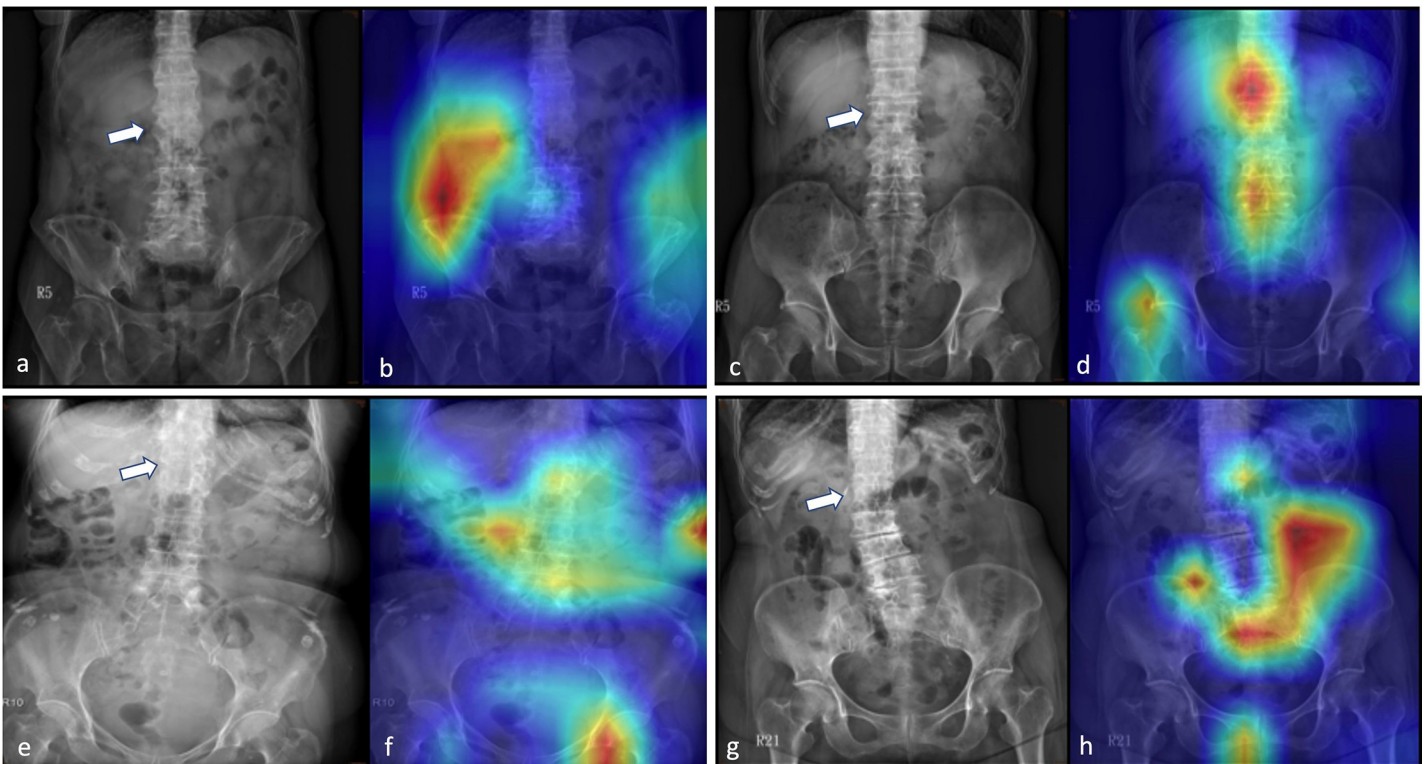

**Fig 4. Grad-CAM-assisted image identification of vertebral fractures.** a,c,e,g: The original plain abdominal frontal radiographs (PARs) with vertebral fractures (arrow); b,d,f,g: Grad-CAM visualized the heatmap images that show that the model tends to focus on regions next to the fracture site.

VFs with morphologic change and spend significant time to double check the accuracy of the diagnosis.

In this study, we performed Grad-CAM to visualize the class-discriminative regions as fracture sites recognized by a DCNN using PARs (Fig 4), similar to the case of normal PARs (Fig 5). An interesting finding is that in PARs without VFs, the heatmaps tend to focus on the spine, and for those with VFs, the focus is on regions next to the fracture site. This can attribute to differences in spine alignment or changes in the soft tissue contrast caused by edema, hemorrhage, or spinal kyphotic deformity. Nonetheless, as mentioned above, the exact characteristics that were used to identify the VFs are not known; for instance, in some images, the activation sites were entirely misidentified, and the reason for this is not clearly understood. In addition, one drawback of the presented system is that it might not be able to identify other diseases because the algorithm was specifically trained to discriminate between spines, with and without VFs.

As future work, we intend to combine ensemble learning and data fusion techniques to achieve a better performance. A web-based system that can input the PARs from PACS has been proposed to create large databases, improve the accuracy of the system, and identify a greater number of undiagnosed patients.

In conclusion, the algorithm trained by a DCNN to identify VFs on PARs showed the potential of delivering a highly accurate and acceptable specific rate and is expected to be useful as a screening tool. To the best of our knowledge, this is the first attempt in which a DCNN was used to identify VFs using PARs. Furthermore, the algorithm could accurately localize the fracture sites to assist physicians in the identification of VFs. This may enhance diagnostic performance and improve consistency in reporting incidentally detected VFs.

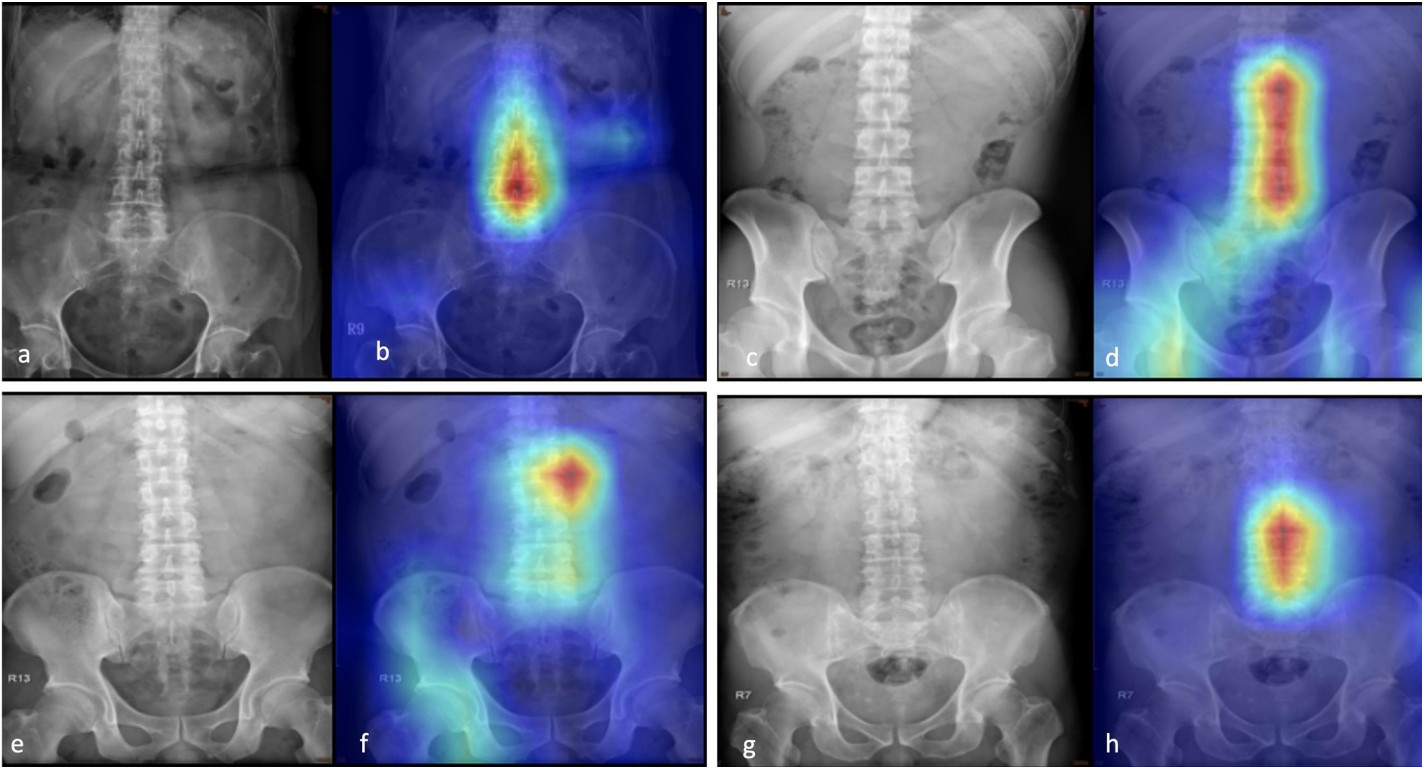

**Fig 5. Grad-CAM-assisted image identification of normal plain abdominal frontal radiographs (PARs).** a,c,e,g: Frontal PARs without vertebral fractures; b,d,f,g: Grad-CAM visualized the heatmap images that show that the model tends to focus on the spine.

## Author Contributions

**Conceptualization:** Hsuan-Yu Chen, Benny Wei-Yun Hsu, Feng-Huei Lin, Tsung-Han Yang, Rong-Sen Yang, Chih-Kuo Lee.

**Data curation:** Hsuan-Yu Chen, Benny Wei-Yun Hsu, Feng-Huei Lin, Rong-Sen Yang.

**Investigation:** Hsuan-Yu Chen.

**Methodology:** Hsuan-Yu Chen, Benny Wei-Yun Hsu.

**Resources:** Tsung-Han Yang, Rong-Sen Yang.

**Software:** Yu-Kai Yin.

**Supervision:** Vincent S. Tseng.

**Validation:** Yu-Kai Yin.

**Writing – original draft:** Hsuan-Yu Chen, Benny Wei-Yun Hsu, Yu-Kai Yin.

**Writing – review & editing:** Hsuan-Yu Chen, Feng-Huei Lin, Rong-Sen Yang, Vincent S. Tseng.

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
