## [Decision Letter · Decision Letter 0]

22 Sep 2020

PONE-D-20-24397

Application of deep learning algorithm to detect and visualize vertebral fractures on plain frontal radiographs

PLOS ONE

Dear Dr. Tseng,

Thank you for submitting your manuscript to PLOS ONE. After careful consideration, we feel that it has merit but does not fully meet PLOS ONE’s publication criteria as it currently stands. Therefore, we invite you to submit a revised version of the manuscript that addresses the points raised during the review process.

From the views of experts, the article presented technique that lacks of any breakthrough that could inspire the researchers of the related field.  Second, about the visualization part, one of the reviewer has the opinion that the heat-map did not indicate the fracture site in Figure 4. and identify the spine in Figure which means the model did not recognize fracture feature. According this point, reviewer doubt the algorithm the authors developed is to recognize spine itself, not fracture part. Third, the authors employ a very small dataset for training and testing which might limited the performance too, Fourth, the authors did not explain well about the method of labelling. Which one is the ground truth of this dataset, this is the most essential elements of deep learning. Hopefully, the author can explain more to clear the doubt of reviewers in the next submission. 

We look forward to receiving your revised manuscript.

Kind regards,

Yan Chai Hum

Academic Editor

PLOS ONE

Journal Requirements:

Reviewers' comments:

Reviewer's Responses to Questions

**Comments to the Author**

1. Is the manuscript technically sound, and do the data support the conclusions?

Reviewer #1: Partly

Reviewer #2: Yes

Reviewer #3: No

2. Has the statistical analysis been performed appropriately and rigorously? 

Reviewer #1: No

Reviewer #2: Yes

Reviewer #3: Yes

3. Have the authors made all data underlying the findings in their manuscript fully available?

Reviewer #1: No

Reviewer #2: No

Reviewer #3: Yes

4. Is the manuscript presented in an intelligible fashion and written in standard English?

Reviewer #1: Yes

Reviewer #2: No

Reviewer #3: No

5. Review Comments to the Author

Reviewer #1: problem statement and project motivation are clearly defined. However, limited explanation on their proposed technique DCNN, why is this neural network been selected? There are many types of CNN are made available for this kind of similar classification works. No benchmarks literature are available to support their proposed techniques. Presently literature are very limited with only 19 references, many state-of-arts neural networks in recent years are not discussed in the present manuscript.

Reviewer #2: 1. Language of the written article is poor. Please recheck it carefully. There are some evidences of weakness in the paper. For example, a number of grammatical errors, including repetitions (of, for, the), there are misspellings/typos in various places. If possible, send the paper for proofreading.

2. The similarity report of the article is high. It should be reduced and some sentences should be corrected in the article.

3. The introduction is a bit stretched and the partition of content between introduction and related work is not appropriate and should be revisited.

4. The paper needs to explain why the work is of significance and applications.

5. The paper needs to compare results with more recent state of the art methods.

6. It would be better to see more recent research papers in references. Also, refer to the paper closely related to your manuscript topic. Please use in your references: “An Efficient Noisy Pixels Detection Model for CT Images using Extreme Learning Machines ".

Reviewer #3: Dear authors,

Thank you and congratulate your study to present the result of DCNN in detecting vertebral fracture with frontal plain film with acceptable performance.

I appreciate your achievement. Indeed, we understand the easily missing diagnosis of vertebral fracture, and with algorithm support, we can help the doctors to make a better diagnosis. Your work might improve current evidence about the DCNN-based algorithm can improve the detection rate of occult fracture.

I have some comments about this manuscript.

First, what is the ground truth of the dataset? The CT finding? Radiologist report? Reviewers’ opinions? When conflict present, how you decide the priority of the definition of fracture.

Second, How about the distribution of fractures in the training set? How about the percentage of occult vertebral fracture in it? And also, how about the percentage of occult fracture in the testing set and validation set? To clarify, this point can make the audience understand the potential possibility of this detection tool.

Third, the visualization image seems not well and cannot accurately identify the fracture's location; how did you label the image? Spot method? Bounding box or other methods. Grad-CAM does sometimes not perform well if the complexity of the dataset and inappropriate hyperparameter selection. Can you describe these to clarify how the visualization produced?

Fourth, from my perspective and experience, using ImageNet as the pretrained dataset might not improve the result and other medical images. If you have other radiograph datasets, use them as the pretrained one will help the transferring learning. There was some literature already discussing this point; please review the state-of-the-art.

Finally, the 70% accuracy is not good enough to prevent further CT or MRI to make the final diagnosis back to the clinical issue.

Thank you again

6. PLOS authors have the option to publish the peer review history of their article (what does this mean?). If published, this will include your full peer review and any attached files.

Reviewer #1: **Yes: **Khin Wee Lai

Reviewer #2: **Yes: **Abidin Caliskan

Reviewer #3: No

---

## [Author Response · Author response to Decision Letter 0]

27 Nov 2020

Response to reviewer

From the views of experts, the article presented technique that lack of any breakthrough that could inspire the researchers of the related field.

Response: Thank you for the comment. Vertebral fractures (VFs) are the most prevalent fractures worldwide but remain largely undiagnosed. In Europe , the economic burden of incident and prior fractures was estimated at € 37 billion and is expected to increase by 25% in 2025. The early diagnosis of VFs is important for medication can reduce the risk of subsequent VFs by 70%, hip fracture by 41%, and other fractures by 25%. We believe that through the automated VFs diagnosis algorithm trained based on the DCNN, which achieved detection of almost double the cases (from 46.6% to 73.59%), enhance diagnostic performance and improve consistency in reporting incidentally detected VFs.

Second, about the visualization part, one of the reviewers has the opinion that the heat-map did not indicate the fracture site in Figure 4. and identify the spine in Figure which means the model did not recognize fracture feature. According this point, reviewer doubt the algorithm the authors developed is to recognize spine itself, not fracture part. 

Response: Thank you for the comment. The Grad-CAM heatmap is a data visualization technique that gives visual cues to the reader about the prediction sources. Our DCNN model can identify the spine in the normal plain film and tend to focus on regions along the fracture site. Our interpretation is that the prediction sources might not contain the VF itself but the differences in the spine alignment or changes in soft tissue contrast due to edema, hemorrhage, or spinal kyphotic deformity, which are caused by VFs.

Third, the authors employ a very small dataset for training and testing which might limited the performance too, 

Response: Use of patient data for secondary purposes is strictly regulated to protect patient privacy, and therefore, in most studies, creation of large datasets is challenging Besides, during dataset training process, we made sure each PARs has CT or MRI images for backup in diagnosis of VFs, we understand if the diagnosis is not correct, it might result in meaningless outcome. That also explained the reason why our dataset was small. However, we use ResNeXt and transfer learning technique to improve our accuracy. A web-based system that can input the PARs from PACS has been proposed to create a larger database in our next study. 

Fourth, the authors did not explain well about the method of labelling. Which one is the ground truth of this dataset, this is the most essential elements of deep learning. 

Response: Thank you for your comment. The PARs datasets were initially labeled as VF or non-VF according to the diagnosis in the registry and supportive images such as CT or MRI were reviewed and reported, respectively, by two experts who were certified by our national board of medical examiners. The final diagnosis was made when the two observers agreed.

Reviewer #1: problem statement and project motivation are clearly defined. However, limited explanation on their proposed technique DCNN, why is this neural network been selected? 

Response: Thank you for the insightful suggestion. However, no CNN-based method that can be applied to all types of image classification problem has yet been developed. Each CNN-based method has its own advantages according to problem definitions and data characteristics. In the various such methods, ResNet [10] has been proven to be one of the most representative deep learning networks. Furthermore, following the architecture of this network, modifications to the architecture of the existing CNN-based methods have been made by utilizing the idea of residual blocks to achieve deeper networks and more efficient learning. In this study, we consider a typical classification method and verify its efficacy. We use ResNeXt [12] as the backbone model for our study. ResNeXt, which is the enhanced network of the ResNet, has shown outstanding results in image classification and has been applied to the field of medical image analysis. The corresponding description has been added in the first paragraph of the section: Development of the algorithm.

There are many types of CNN are made available for this kind of similar classification works. No benchmarks literature is available to support their proposed techniques.

Response: Thank you for the suggestions. We use ResNeXt for this study because ResNeXt has been successfully applied to medical application (“Ambulatory Atrial Fibrillation Monitoring Using Wearable Photoplethysmography with Deep Learning”, KDD 2019). We have added the related paper in our reference.

Presently literature are very limited with only 19 references, many state-of-arts neural networks in recent years are not discussed in the present manuscript.

Response: Thank you for your comment. We have added more references and more discussions of neural networks in the section: Development of the algorithm.

Reviewer #2: 

1. Language of the written article is poor. Please recheck it carefully. There are some evidences of weakness in the paper. For example, a number of grammatical errors, including repetitions (of, for, the), there are misspellings/typos in various places. If possible, send the paper for proofreading.

Response: Thank you for the suggestion. We availed the services of a English editing company to revise the manuscript for improved grammar, clarity, and readability.

2. The similarity report of the article is high. It should be reduced and some sentences should be corrected in the article.

Response: Thank you for the comment. We have revised the article to eliminate the existing redundancy and we also engaged a professional editing company help us in this regard.

3. The introduction is a bit stretched and the partition of content between introduction and related work is not appropriate and should be revisited.

Response: We really appreciate the reviewers’ effort and criticism to make this manuscript better. We have revised the manuscript accordingly.

4. The paper needs to explain why the work is of significance and applications.

Response: Thank you for your comment. Vertebral fractures (VFs) are the most prevalent fractures in the world and remain largely undiagnosed. The economic burden of incident and prior fractures was estimated at € 37 billion and is expected to increase by 25% in 2025 in Europe. The early diagnosis of VFs is important for medication can reduce risk of subsequent VFs by 70%, hip fracture by 41%, and other fractures by 25%. We believe through the automated VFs diagnosis algorithm trained based on the DCNN, which almost double identified cases (from 46.6% to 73.59%), may enhance diagnostic performance and improve consistency in reporting incidentally detected VFs.

5. The paper needs to compare results with more recent state of the art methods.

Response: Thank you for the suggestion. Plain abdominal frontal radiographs (PARs) are a common investigation method performed for a variety of clinical conditions, such as urinary or gastrointestinal disorders. However, it is not easy for physicians to screen for VFs based on PARs, as general physicians typically focus on their own specialties. To the best of our knowledge, this is the first attempt using DCNN to identify VFs with PARs. 

6. It would be better to see more recent research papers in references. Also, refer to the paper closely related to your manuscript topic. Please use in your references: “An Efficient Noisy Pixels Detection Model for CT Images using Extreme Learning Machines ".

Response: Thank you for the suggestion. We have added the paper: “An Efficient Noisy Pixels Detection Model for CT Images using Extreme Learning Machines " in our reference in the introduction section of this manuscript. Each CNN-based method has its own advantages according to problem definitions and data characteristics. ResNet is one of the most representative deep learning methods that brings innovation to the architecture of CNN for deeper networks and efficient learning. In this study, we consider not only the classical method but the effectiveness of classification. Therefore, we use ResNeXt as the backbone model for this study. ResNeXt has achieved success in the applications of medical image analysis. The corresponding description has been added in the section: Development of the algorithm.

Reviewer #3: Dear authors, Thank you and congratulate your study to present the result of DCNN in detecting vertebral fracture with frontal plain film with acceptable performance. I appreciate your achievement. Indeed, we understand the easily missing diagnosis of vertebral fracture, and with algorithm support, we can help the doctors to make a better diagnosis. Your work might improve current evidence about the DCNN-based algorithm can improve the detection rate of occult fracture. I have some comments about this manuscript.

First, what is the ground truth of the dataset? The CT finding? Radiologist report? Reviewers’ opinions? When conflict present, how you decide the priority of the definition of fracture.

Response: Thank you for your encouraging comment. The PARs datasets were initially labeled as VF or non-VF according to the diagnosis in the registry and supportive images such as CT or MRI were reviewed and reported, respectively, by two experts who were certified by our national board of medical examiners. The final diagnosis was made when the two observers agreed. 

Second, How about the distribution of fractures in the training set? How about the percentage of occult vertebral fracture in it? And also, how about the percentage of occult fracture in the testing set and validation set? To clarify, this point can make the audience understand the potential possibility of this detection tool.

Response: Thank you for the valuable suggestion. The PARs dataset was split into the training data (80%) and validation data (20%). To ensure reliability of our experiments, we applied 5-fold cross validation during training. For each set, the proportion of the fracture images and normal images was kept equal.

Third, the visualization image seems not well and cannot accurately identify the fracture's location; how did you label the image? Spot method? Bounding box or other methods. Grad-CAM does sometimes not perform well if the complexity of the dataset and inappropriate hyperparameter selection. Can you describe these to clarify how the visualization produced?

Response: Thank you for the suggestion. The Grad-CAM heatmap is an assisted data visualization technique that gives visual cues to the reader about the prediction sources. Our algorithm trained based on the DCNN can identify the spine in normal plain film and tend to focus on regions along the fracture site. Our interpretation is that the prediction sources might not be the VF itself but the differences in spine alignment or changes in soft tissue contrast due to edema, hemorrhage, or spinal kyphotic deformity which were caused by the VFs.

Fourth, from my perspective and experience, using ImageNet as the pretrained dataset might not improve the result and other medical images. If you have other radiograph datasets, use them as the pretrained one will help the transferring learning. There was some literature already discussing this point; please review the state-of-the-art.

Response: Thank you for the suggestion. Using patient data for secondary purposes was under strict regulation to protect patient’s privacy, so in most studies it was hard to create huge dataset. Therefore, we use transfer learning technique to improve our accuracy. Although ImageNet is not a medical specific dataset, it can still help us obtain better initial parameters of the network for further learning process (“Pre-training on Grayscale ImageNet Improves Medical Image Classification”, ECCV 2018). The corresponding description has been added in the section: Development of the algorithm. A web-based system that can input the PARs from PACS has been proposed to create a larger database in our future study.

Finally, the 70% accuracy is not good enough to prevent further CT or MRI to make the final diagnosis back to the clinical issue.

Response: Thank you for your comment. Vertebral fractures (VFs) are the most prevalent fractures and remain largely undiagnosed. We believe that through the use of automated VFs diagnosis algorithm trained based on the DCNN, almost double the cases (from 46.6% to 73.59%) can be identified, which may enhance the diagnostic performance and improve consistency in reporting incidentally detected VFs. We aimed at screening the neglected VFs and increase the plain film of the spine lateral view, which can be helpful in the diagnosis of suspicious cases (accuracy of more than 90%). Based on the results, the doctors can then decide if patients need advanced CT or MRI examinations, which is expensive.

---

## [Decision Letter · Decision Letter 1]

12 Jan 2021

Application of deep learning algorithm to detect and visualize vertebral fractures on plain frontal radiographs

PONE-D-20-24397R1

Dear Dr. Tseng,

We’re pleased to inform you that your manuscript has been judged scientifically suitable for publication and will be formally accepted for publication once it meets all outstanding technical requirements.

Kind regards,

Yan Chai Hum

Academic Editor

PLOS ONE

Additional Editor Comments (optional):

Reviewers' comments:

Reviewer's Responses to Questions

**Comments to the Author**

1. If the authors have adequately addressed your comments raised in a previous round of review and you feel that this manuscript is now acceptable for publication, you may indicate that here to bypass the “Comments to the Author” section, enter your conflict of interest statement in the “Confidential to Editor” section, and submit your "Accept" recommendation.

Reviewer #1: All comments have been addressed

Reviewer #2: All comments have been addressed

2. Is the manuscript technically sound, and do the data support the conclusions?

Reviewer #1: Yes

Reviewer #2: Yes

3. Has the statistical analysis been performed appropriately and rigorously? 

Reviewer #1: Yes

Reviewer #2: Yes

4. Have the authors made all data underlying the findings in their manuscript fully available?

Reviewer #1: Yes

Reviewer #2: Yes

5. Is the manuscript presented in an intelligible fashion and written in standard English?

Reviewer #1: Yes

Reviewer #2: Yes

6. Review Comments to the Author

Reviewer #1: based on the authors revision, and the point to point reviewers responses, all comments seems addressed satisfactory, recommend for acceptance

Reviewer #2: The manuscript shows clear approach to the topic, the methodology and the conclusions.

The paper results compared with recent state of the art methods.

Recent research papers are addressed in references.

7. PLOS authors have the option to publish the peer review history of their article (what does this mean?). If published, this will include your full peer review and any attached files.

Reviewer #1: **Yes: **Lai Khin Wee

Reviewer #2: No

---

## [Editor Report · Acceptance letter]

18 Jan 2021

PONE-D-20-24397R1 

Application of deep learning algorithm to detect and visualize vertebral fractures on plain frontal radiographs 

Dear Dr. Tseng:

I'm pleased to inform you that your manuscript has been deemed suitable for publication in PLOS ONE. Congratulations! Your manuscript is now with our production department. 

Kind regards, 

on behalf of

Dr. Yan Chai Hum 

Academic Editor

PLOS ONE